A survey of researchers’ code sharing and code reuse practices, and assessment of interactive notebook prototypes

Cadwallader Lauren lcadwallader@plos.org
Hrynaszkiewicz Iain
Public Library of Science , Cambridge , United Kingdom
Brembs Björn
Electronic publication date: 2022 Aug 22
Publication date: 2022
Volume: 10
Electronic Location ID: e13933
Received 2022 Jun 7; Accepted 2022 Aug 1
Copyright: ©2022 Cadwallader and Hrynaszkiewicz
Copyright year: 2022
Copyright holder: Cadwallader and Hrynaszkiewicz
License: This is an open access article distributed under the terms of the Creative Commons Attribution License, which permits unrestricted use, distribution, reproduction and adaptation in any medium and for any purpose provided that it is properly attributed. For attribution, the original author(s), title, publication source (PeerJ) and either DOI or URL of the article must be cited.
License URL: https://creativecommons.org/licenses/by/4.0/

Keywords: Open science, Publishing practices, Research code dissemination, Research code reuse, Research code sharing, Survey results

Funding: The authors received no funding for this work.

==============================
This research aimed to understand the needs and habits of researchers in relation to code sharing and reuse; gather feedback on prototype code notebooks created by NeuroLibre; and help determine strategies that publishers could use to increase code sharing. We surveyed 188 researchers in computational biology. Respondents were asked about how often and why they look at code, which methods of accessing code they find useful and why, what aspects of code sharing are important to them, and how satisfied they are with their ability to complete these tasks. Respondents were asked to look at a prototype code notebook and give feedback on its features. Respondents were also asked how much time they spent preparing code and if they would be willing to increase this to use a code sharing tool, such as a notebook. As a reader of research articles the most common reason (70%) for looking at code was to gain a better understanding of the article. The most commonly encountered method for code sharing–linking articles to a code repository–was also the most useful method of accessing code from the reader’s perspective. As authors, the respondents were largely satisfied with their ability to carry out tasks related to code sharing. The most important of these tasks were ensuring that the code was running in the correct environment, and sharing code with good documentation. The average researcher, according to our results, is unwilling to incur additional costs (in time, effort or expenditure) that are currently needed to use code sharing tools alongside a publication. We infer this means we need different models for funding and producing interactive or executable research outputs if they are to reach a large number of researchers. For the purpose of increasing the amount of code shared by authors, PLOS Computational Biology is, as a result, focusing on policy rather than tools.

Introduction

Code sharing requirements of journals and funders are increasing but are not as prevalent as requirements for sharing other research outputs, such as research data. Software tools, such as code notebooks, can facilitate code sharing in a way that reduces barriers to computational reproducibility but are not necessarily cost (e.g., time) free to authors. Some publishers have experimented with executable code and interactive features in their articles. Policies can also be employed to increase the amount of code shared alongside published articles. Researchers working in fields such as computational biology generate code for a large proportion of their studies (Hrynaszkiewicz, Harney & Cadwallader, 2021a; Hrynaszkiewicz, Harney & Cadwallader, 2021b). Sharing code improves reproducibility, especially when made available before publication (Fernández-Juricic, 2021). Lack of source code—along with raw data, and protocols—has been described as the main barrier to computational reproducibility of published research (Seibold et al., 2021). However, technical and cultural barriers to computational reproducibility have been identified in the literature (Samota & Davey, 2021; Hrynaszkiewicz, Harney & Cadwallader, 2021a; Van den Eynden et al., 2016). These barriers include insufficient time, funds and skills to prepare code for sharing. A desire to protect intellectual property (IP) is also reported as a common or important barrier to code sharing.

Journals and publishers must understand and respond to these challenges in the research communities they serve if they wish to support open, reproducible research, and test and implement solutions. Introducing policies is an important way for journals to increase awareness and adoption of research practices that are important to a particular community, as demonstrated by the increase in research data sharing policies and practices in the last decade (Hrynaszkiewicz, 2020). In 2021, PLOS Computational Biology introduced a strengthened, mandatory code sharing policy in response to a desire of this community to support reproducibility by increasing the availability of code associated with articles published in the journal (Cadwallader et al., 2021). The introduction of this policy was supported by the results of a survey of the computational biology community, which demonstrated their support for a mandatory code sharing policy in PLOS Computational Biology (Hrynaszkiewicz, Harney & Cadwallader, 2021a). The survey results also found that code sharing and access are important to researchers, and that they are satisfied with their ability to share their own code, but they are not satisfied with their ability to access other researchers’ code. Following the Jobs To Be Done theory (Ulwick & Osterwalder, 2016), this finding implies that there may be opportunities for new solutions (which could be products, policies, services or features) that support researchers in accessing other researchers’ code.

Numerous technical solutions (tools) exist that could play a role in improving code availability, and reuse. Scholarly publishers and tool providers have experimented with interactive and reproducible articles for years (Akhlaghi et al., 2021). Such tools inherently require availability of code and data to enable interactivity with and reuse of results. An example of this is the journal, eLife, and reproducible document platform, Stencila, who have collaborated to experiment with publication of Executable Research Articles (ERA; Tsang & Maciocci, 2020). Other tools that support code sharing and reuse alongside scholarly articles include commercial platforms such as Code Ocean, which provides executable code capsules; Gigantum, and NextJournal (Perkel, 2019) and collaborative, interactive code notebooks such as Observable (Perkel, 2021). For a review of infrastructures that support computational reproducibility see Konkol, Nüst & Goulier (2020). Many code notebook tools are built on open source technology, such as Jupyter and MyBinder, and researcher-led efforts to produce code notebook type outputs often use these (Lasser, 2020). One relatively new code notebook initiative, NeuroLibre, supported by the Canadian Open Neuroscience Platform, is an open access platform hosting notebooks derived from published or preprinted research articles that can be freely modified and re-executed (Boudreau et al., 2021).

The potential benefits of these tools—for researchers as readers and authors, for publishers, and the accessibility of science—are numerous. Our focus was on how these tools meet researcher needs for code sharing and reuse, as these needs align with PLOS’ goals to increase the adoption, and benefits, of open science. But the extent to which these tools do meet these needs is unclear from the available literature. Furthermore, the adoption of new tools or workflows for preparing and sharing code would incur costs, in terms of time and effort, for researchers (as authors, readers, editors and peer reviewers) and publishers. For new tools to be widely adopted it is important to understand if additional effort required to adopt new tools is acceptable to their users. As a publisher PLOS experiments with solutions that support open science in different communities, and partners with community resources, such as data repositories and preprint servers, to achieve this. To this end, rather than creating new solutions, PLOS partnered with NeuroLibre to learn more about the value of their interactive code notebooks and research publications to readers and authors. The results were anticipated to:

– Provide a deeper understanding of how researchers share and interact with code

– Inform PLOS Computational Biology’s plans for further supporting code sharing and reuse, beyond its mandatory code sharing policy

– Inform development of NeuroLibre with quantifiable feedback from potential users of the tool on the tool itself and researcher needs that are related to the features of the tool

– Provide PLOS, and other publishers, with quantitative insights on researchers’ attitudes and experience with interactive article features, to inform future publishing innovation approaches.

Methods

We created a survey in English in Alchemer and distributed it in February and March 2021. The survey had three main purposes:

(1) Understand how researchers interact with code as readers of articles

(2) Gather feedback on the prototype NeuroLibre notebook version of PLOS Computational Biology articles

(3) Gain a more detailed understanding of researchers’ abilities to carry out code sharing tasks, how they rate the importance of these tasks and how satisfied they are with their ability to complete the tasks

The survey was promoted with an accompanying blog (Cadwallader, 2021) and email campaign, which was sent to previous PLOS authors and other PLOS registered users in computational biology related disciplines (n = 23, 272). The survey (Cadwallader, Hrynaszkiewicz & Harney, 2022) was launched with the blog on the 11th February 2021 and the email campaign followed on the 19th February. The results were exported from Alchemer on 25th March 2021.

The survey methodology was adapted from our group’s previous recent work (described in Hrynaszkiewicz, Harney & Cadwallader, 2021b). Briefly, respondents were asked to answer a series of questions from the perspective of both readers and authors of articles with associated code. To identify if there were opportunities to support researchers with sharing code using new solutions, we asked respondents to rate various code sharing and reuse factors in terms of how important they were to them and how satisfied they were with their ability to complete them. These responses were converted to numerical scores and used to calculate opportunity scores for each factor using the following equation: Opportunity score=Mean importance∗1−mean satisfaction/100.

Opportunity scores above 25 indicate “better than neutral” or marginal opportunities and scores above 36 we regard as good opportunities. This approach is more nuanced than simply using quadrants and looking for high importance/low satisfaction scores.

In addition, NeuroLibre created two prototype interactive notebook versions (Larremore, 2019; Tampuu et al., 2019) of articles published in PLOS Computational Biology (Larremore, 2019; Tampuu et al., 2019), so they could be shared with the community and their feedback sought on the value and features of the interactive format. Survey respondents were asked to give feedback on one of these prototypes.

Ethical considerations

Approval from a research ethics committee was not sought as we considered the research to be low risk. Sensitive information about the participants was not collected and all data were collected anonymously. Participants were informed that participation was voluntary, and that they were free to withdraw at any time until they submitted their response. The results were only analyzed in aggregate and answers were never associated with individual participants. The data collection procedures and survey tool are compliant with the General Data Protection Regulation 2016/679.

Results

Respondent demographics

The survey received a total of 188 complete responses, with an additional 39 partial responses (some but not all questions answered) and 175 incomplete responses (some but not all demographic questions answered only). 79% of the respondents clicked through from the email campaign link (n = 316), which had a 1.4% engagement (click) rate. This analysis will focus on the 188 complete responses.

A range of disciplines are represented by the respondents, with a third of respondents being from the computational biology field (Table 1). For those who chose ‘Other’, 13 out of 14 respondents were in STEM fields, with math-related fields being most commonly specified (n = 6). One individual was from a social sciences discipline.

Table 1 Disciplinary distribution of respondents who completed the survey (n = 188).

Research field	n	% of total	Research field	n	% of total	
Computational Biology	63	34%	Engineering and Technology	9	5%	
Biology and Life Sciences	34	18%	Physical sciences	10	5%	
Bioinformatics	32	17%	Ecology and Environmental Sciences	5	3%	
Medicine and Health Sciences	18	10%	Social sciences	3	2%	
Other (please specify)	14	7%				

Responses are skewed more towards researchers with fewer publications, (Fig. 1). Respondents were overwhelmingly from Europe (46%) or North America (40%), with very few respondents indicating their location in other geographic regions (Table 2). 54% of respondents had previously published in PLOS Computational Biology.

Figure 1 Distribution of respondents (n = 188) according to the number of previously published papers.

The number of respondents in each category is given above the bar.

Table 2 Geographical distribution of respondents (n = 188).

Region	n	% of total cohort	
Europe	87	46%	
North America	75	40%	
Asia	12	6%	
South America	7	4%	
Australasia	4	2%	
Africa	2	1%	
Middle East	1	1%	

When and why researchers access or read code

Respondents were asked to answer a set of questions from the viewpoint of a reader of research articles that had associated code to understand how they interacted with code in this setting. Three-quarters (n = 141) of the respondents look at code associated with a research paper at least occasionally, with 39% (n = 74) looking at code frequently or very frequently. Only 6% (n = 12) said they never looked at the associated code (Fig. 2).

Figure 2 Frequency with which respondents look at code associated with research articles.

The number of respondents is given above each bar.

The degree to which readers from different disciplines look at code associated with research articles is variable, although many of the cohorts included in the survey results are small (Fig. 3). Of the largest cohorts surveyed, those in the Biology and Life Sciences look at code associated with articles less frequently than in Computational Biology and Bioinformatics. Lower levels of looking at code are also seen in the Medicine and Health Sciences cohort although this is a smaller group (n = 18).

Figure 3 Frequency at which authors look at code associated with research papers according to discipline.

Respondents were asked why they look at code associated with published articles. Free text answers were provided by 178 respondents. Answers were categorised to identify general trends, with the majority of respondents (n = 100) giving two or more reasons for looking at the code.

• 125 (70%) respondents look at code to aid their understanding of the article. For example, 113 respondents (63%) specified that they wish to directly verify the code or examine its use in the context of the research presented and 38 respondents (21%) look at the code to better understand the methods described in the article, e.g., what parameters were selected.

• 86 (48%) respondents gave answers that fell into the ‘reuse’ category, e.g., directly reusing the code (62 responses/35%) and reusing selected parts of the code (27 responses/15%). Other reuse reasons were using the code as an example in teaching (one response), as a comparison to the reader’s own code (six response/3%) and to reuse the data (one response).

• Respondents also looked at the code to assess the quality of the research (37 respondents/21%), giving reasons such as to check for minimal standards (eight responses/4%), for trust or transparency reasons (five responses/3%) and replicate the analysis using their own data (21 responses/12%).

• Reasons linked to discovery were also given by five respondents (3%), for example finding new GitHub repositories of interest and looking for novel code.

The usefulness of methods for accessing or reading code

Respondents were asked how useful they found various methods of accessing code associated with a research article, when considering the 6 months before they completed the survey. Not all respondents had encountered the methods specified. Using a ‘Link to a code repository’ was the most common method (encountered by 98%), followed by ‘link to a website’ (88%) and ‘available on request’ (87%) (Fig. 4). A link to archived code, that is, a snapshot of code deposited in a generalist repository was encountered by 72% of respondents. Links to code notebooks were encountered by 66% and executable code capsules by 40%. The methods were not defined for respondents, although they had been asked to look at a prototype notebook before answering the questions.

Figure 4 Rates that the various methods of code sharing have been encountered by the respondents.

‘Link to code repository’ was rated as the most useful method –both in terms of the number of respondents who rated it ‘extremely’ or ‘very useful’, and the number who rated it as ‘not at all useful’ (Fig. 5). Accessing code that is ‘available on request’ was rated as least useful (based on number of ‘not at all useful’).

Figure 5 The usefulness of different code sharing methods as percentages of the respondents who had encountered each method.

The five-point unipolar scale used in this question can be mapped to a value from 0 to 100, with 0 equalling ‘not at all useful’ and 100 equalling ‘extremely useful’. ‘I have not encountered this method of sharing’ responses were not scored. Taking the mean rating for all the methods (Fig. 6), the most commonly encountered method (link to a code repository), is also the most useful (mean 84.1 ± 3.2 (95% CI)). The mean scores given were: code notebooks (69.9 ± 5.3 (95% CI)); link to archived code (64.5 ± 4.4 (95% CI)); link to website (52.3 ± 4.2 (95% CI)); and executable code capsules (50.8 ± 8.9 (95% CI)). The 95% confidence intervals for code capsules and link to a website (41.9–59.7 and 48.1–56.5 respectively) do not overlap those for code notebooks and archived code (64.6–75.1 and 60.0–68.9 respectively).

Figure 6 Mean usefulness of different methods of code sharing.

A score of zero equates to ‘not at all useful’ and 100 equals ‘extremely useful’. The mean values are given above the bars. Error bars show the 95% confidence interval.

The reasons why researchers favoured certain methods of accessing code were gathered via a free text question. The most common reasons, which all received between 18 and 10 mentions, were (in order of number of mentions):

– Ability to see new versions of the code (most associated with code repositories1)

– Quick to access the code (most associated with code repositories)

– The method allows exploration of the code, which aids understanding (most associated with notebooks)

– The method is associated with good documentation/README files (most associated with code repositories)

– The practicality of the method (most associated with code repositories)

– The method provides long term access to the code (most associated with archived code2)

– The method allows for reproduction of results (most associated with code repositories and notebooks)

– It is an established method (most associated with code repositories)

Features of code notebooks that are useful when accessing or reading code

All respondents were then asked to rate the importance of various features of the NeuroLibre prototype notebook (Larremore, 2019) using a 5-point unipolar scale, or selected that they did not use the feature. Converting these responses to numerical scores on a scale of 0 to 100 and taking the mean (Table 3) gives us a sense of the features readers value the most. The top two features—‘having all the code, data and figures in one place’ and ‘knowing the code is running in the right environment’—are not features unique to code notebooks. Features related to the interactivity elements of the notebook, e.g., ability to change parameters of the figures, had mean scores in the low to mid 60s. The lowest scoring feature was ‘having extra figures included that were not in the original paper’.

Table 3 Mean importance scores given to each feature of the notebook.

Table excludes those who answered they did not use the feature or were not aware of its presence/absence. The higher the score, the more important the feature is.

	Mean score	stdev	n	
Having all the code, data and figures in one place	81.0	22.7	186	
Knowing the code is running in the right environment*	73.5	29.3	178	
Ability to interact inline with the code in the browser**	66.6	28.5	178	
Ability to uncover the data point by hovering over the points on the graph*	65.8	25.3	184	
Ability to open up the code as a Jupyter notebook*	64.9	30.5	173	
Ability to zoom in/out on the figures*	63.3	26.5	182	
Ability to change the parameters of the figure*	62.6	26.7	185	
Having extra figures included that were not in the original paper*	53.8	29.0	185	
Notes.

* Features marked were included in the NeuroLibre prototype.

** Features marked are present in the NeuroLibre prototype but were not working during the survey period.

Importance and satisfaction of factors associated with sharing code from an author’s perspective

Importance and satisfaction responses were converted to numerical scores as described in the Methods section. All factors scored above 50 for mean importance, with standard deviations ranging between 20.6 and 33.3 (Table 4 and Fig. 7). ‘Ability to share my code with good accompanying documentation’ received the highest mean importance score (82.2, SD: 20.6) and was also fairly well satisfied (72.2 , SD: 23.2). All of the factors have a mean satisfaction score above 50, although the standard deviations all range between 23.2 and 28.8. The lowest scoring factors are ‘Readers can easily run the code in the correct environment’ (mean satisfaction score 55.4 , SD: 28.0) and ‘The data and code are in the same place’ (mean satisfaction score 60.4 , SD: 28.8). These are both considered important factors (means scores 76.1 , SD: 23.8 and 73.0 , SD: 28.0 respectively). These are the only two factors that have an opportunity score above 25, although they are not above 36, and therefore present only a marginal opportunity.

Time spent on preparing code as authors

The survey also asked questions about the amount of time authors spent preparing to share their code. The majority of respondents spend more than one day preparing code and this observation holds true when it is separated into cohorts based on the number of papers published (Fig. 8). The researchers with the most papers (>50) are most likely to take more than one week to prepare their code for sharing, whereas the most common response for researchers with fewer papers (<50) was more than one day but less than one week. This may be a reflection on the number of additional constraints on time felt by more established, i.e., published, researchers, such as teaching or supervision of students.

Time authors are willing to spend improving their methods of sharing code

Respondents were also asked how much extra time they would be willing to spend on using a new tool to make the code easier to read and run. This question was chosen as our preliminary interviews with researchers suggested that making code easier to run and read for others was important for authors, which is supported by the satisfaction and importance scores seen in this survey (Table 4 and Fig. 7). Answers were varied, with the top three responses being ‘more than one day’ (36%), ‘a day’ (21%) and ‘a couple of hours’ (20%). There does not appear to be a trend if the respondents are split into cohorts based on the number of previous publications (Fig. 9). However, those who already spend more than a day preparing their code are more likely to spend extra time on a new tool to improve their code.

Table 4 Mean satisfaction and importance scores for code sharing factors.

Respondents were asked to rate the factors using a 7 and 5 point Likert scale respectively, which have been converted into scores out of 100. The higher the score, the more important or satisfied the respondent is. The final factor in the table was only asked in relation to importance.

	Mean Satisfaction	Standard deviation	n	Mean importance	Standard deviation	n	Opportunity score	
Ability to share the code in my preferred form, e.g., as zip file or executable code capsule	68.6	26.3	165	60.7	28.3	182	19.1	
Ability to share the code in my preferred location/platform	74.0	23.3	176	68.7	29.1	187	17.9	
Ability to share my code with good accompanying documentation	72.2	23.2	178	82.2	20.6	187	22.8	
Spend less time on the preparing my code for sharing, e.g., cleaning code, writing documentation	67.1	24.3	177	62.5	31.6	188	20.6	
Spend less time on uploading the code and documentation to a repository	75.9	23.7	179	56.5	33.3	188	13.6	
Depositing my code in a permanent archive	71.1	26.2	169	74.5	26.9	187	21.5	
Readers can easily run the code in the correct environment	55.4	28.0	171	76.1	23.8	186	33.9	
The data and code are in the same place	60.4	28.8	172	73.0	28.0	183	28.9	
Curation checks are run on my code by a third party				58.9	29.4	176		

Discussion

What do readers value and why?

The findings from this survey show the most prevalent reason for readers looking at code was for verification or examination purposes, with 70% of respondents looking at the code to aid their understanding of the article. In journals where word limits apply, the reproducibility of the research can be compromised if methodological details—in this case computational methods—are not fully detailed (Samota & Davey, 2021; Haddaway & Verhoeven, 2015) and it is unsurprising, therefore, that researchers commonly look at code to aid their understanding of the work. The number of respondents who wished to rerun (rather than examine) the code for reproducibility reasons was lower (∼16%), which has also been observed in other studies (Peterson & Panofsky, 2021).

Figure 7 Plot of mean satisfaction and importance scores for each factor related to being an author of code.

The blue shaded areas denote where factors have to plot in order to be considered marginal or good opportunities using the Opportunity Score as outlined in the Methods section. Data for the figure are given in Table 4.

Figure 8 Amount of time spent preparing code for publication by number of published papers.

Figure 9 Amount of time researchers are willing to spend using a new tool to make their code easier to read and run, by number of published papers.

The desire to look at the code rather than run it aligns well with the ranking of a code repository, such as Github, as the most useful method for accessing code by readers (only 1% ranked it as not at all useful), as the presentation of code in these repositories lends itself to exploration or examination but not to immediately rerunning or interacting with code. This survey did not map participants’ workflows so they could be downloading and running code locally, although this is not always easy or possible (Samota & Davey, 2021). 98% of respondents had encountered code shared via code repositories and this prevalence is perhaps a factor in its high usefulness scores as it is widely used by researchers in computational disciplines. The high encounter rate combined with the high usefulness scores indicates that generally readers are satisfied with the most common methods of code sharing.

The survey results also show best practice for code sharing (depositing code in an archive repository) has been encountered by 72% of our respondents. This is a higher percentage than seen in our previous research on data sharing practices where 56% deposit data in a repository. With both code and data, often researchers aren’t following what is considered to be best practice (using repositories) but are satisfied with their ability to share data, from their perspective (Hrynaszkiewicz, Harney & Cadwallader, 2021b).

At the other end of the scale (discounting the “available on request” option which was viewed very negatively), executable code capsules had the lowest mean usefulness score of all the methods presented (50.8) whereas code notebooks scored higher (69.9). This is interesting given that they have similar features and aims and raises the question: what are notebooks doing better than code capsules, or what needs are they meeting that capsules aren’t? Unfortunately, we cannot answer that question directly with our survey data.

The survey question on why readers favoured certain methods of access give some insight into user needs when it comes to accessing code. Versioning, good documentation and long term access are elements considered best practice for code sharing (Lamprecht et al., 2020) and were all amongst the most common reasons given for preferred methods. The other reasons relate to what readers wish to do with the code—explore the code and/or reproduce the results in a quick and accessible manner—and are what these methods of code sharing are good at facilitating.

Prototype notebook features

Respondents were asked to rank the importance of a range of features they may have encountered in the prototype notebook, however, many of these features are not exclusive to this notebook and can be found in other code sharing tools. Presenting the prototype notebooks may have affected the respondents’ answers to the usefulness of the features, however, given that a third of respondents had not encountered a notebook associated with a research article in the last 6 months the prototype did offer some useful context to those participants and gave all respondents a similar experience to guide their answers. Readers scored ‘having all the code, data and figures in one place’—a feature also present in tools such as code capsules—as the most important (mean score 81.0/100; see Table 3). The usefulness of having code, data and figures in one place aligns with how information is often presented in a published article: figures are together with the text, and the data and code are shared (if they are shared) on a different, or multiple different, platforms making the research outputs dispersed. This issue could be solved in a number of different ways, either through technological solutions (such as notebooks, executable code capsules or imbedded repository widgets on article pages), publishing practices (such as requiring authors to share outputs in a certain way) or through changing researcher behaviour so they share their research as a single package of text, figures, data and code regardless of any mandates or policies they have to comply with or solutions offered by publishers.

The second highest scoring feature (mean score 73.5/100) was ‘knowing the code is running in the right environment’. Samota & Davey (2021) found that even researchers trained in computational methods had regularly encountered technical barriers to computational reproducibility. Containerisation—packaging the code and all the components needed to run it correctly—is one solution to this problem. It is interesting that this factor scores so high, yet so few respondents wish to run the code, or rated solutions, such as notebooks and executable code capsules, highly for usefulness. Authors scored their satisfaction with their ability to ensure readers are running their code in the correct environment the lowest out of all factors we surveyed (mean 55.4, SD: 28.0). Although this is the lowest score, it is still above 50 and so there is little opportunity to better support this activity. It is not clear from our survey findings that offering a tool to assist with readers running their code in the correct environment would meaningfully change the way readers interact with code although perhaps the possibility of verifying reproducibility will increase confidence in the results (Nosek et al., 2015).

The ability to interact with the code inline was ranked as the third most important feature of the prototype code notebook, which supports readers’ desire to run, and possibly modify, the code in the correct environment. Conversely, Samota & Davey (2021) found a “link to the source code of interactive figures” the least valued feature out of the list in the survey. While this may suggest that readers don’t wish to run the code, it may also be an indication that readers don’t like having to access links to code (contrary to our findings that researchers like accessing code via repositories). The interactive features, such as zooming in on data points or changing parameters, had lower importance scores, in the low to mid 60s, falling between the moderately important (50/100) and very important (75/100) rating. No one feature of the notebook stands out as being the main reason why respondents would look at a notebook like the one tested—those who scored the likelihood of looking at the notebook highly, generally scored each of the features highly as well.

Other opportunities to support authors

Authors’ ability to share the code with good documentation had the highest mean importance score (82.2, SD: 20.6) and a high satisfaction score (mean 72.2, SD:23.2) and good documentation was commonly given as a reason by readers for their preferred method of accessing data. In another survey of computational biology authors (Hrynaszkiewicz, Harney & Cadwallader, 2021a), we found that there was a disconnect between how satisfied researchers are with their ability to share code well and the ability of others to share code. That data suggest authors regard themselves as competent at this task but view the competence of others less favourably. This is an area of interest that is worth future exploration to understand if this perceived gap in skills is genuine.

Comparing policy to technology as solutions for increasing code sharing

There is evidence from our survey and others (e.g., Perkel, 2017; Samota & Davey, 2021) that researchers regard the ability to interact with code published in its complete software environment as beneficial. Using containerisation tools, such as Docker, have been recommended for increasing the reproducibility of research (Burton et al., 2020) but it has also been acknowledged that this requires skills that not many researchers in this field have (Kim, Poline & Dumas, 2018). Platforms that utilise this technology have been adopted or trialled by several publishers, for example Code Ocean has been deployed by some Springer Nature journals, and some Taylor & Francis journals.

However, it has been acknowledged that authors already using GitHub and Zenodo may feel that the creation of a code capsule is redundant (Cheifet, 2021). The trial of code capsules at several Nature journals demonstrated that peer reviewers were verifying the code and reproducing the results of the manuscripts they were assessing (Cheifet, 2021) but it is unclear to what extent this was above the level of reviewer engagement seen before the trial or what proportion of reviewers were engaging in this type of activity. Our survey was focused on the needs of readers and authors rather than peer reviewers, but showed that readers have mixed feelings about the usefulness of executable code capsules.

Samota & Davey (2021) state that top-down requirements from journals to release reproducible data and code will in part rely on the availability of technical solutions that are accessible and useful to most scientists. In one sense, these solutions are already available in the form of code repositories, although we acknowledge this doesn’t enforce reproducible code and data sharing because the code is not curated or reviewed. However, technology is only one barrier and the journals that have implemented enhanced solutions are, to our knowledge, yet to show that these are making a significant difference to the quality or amount of code that is shared. Additionally, the added benefit, as opposed to the perceived benefit, that they bring to authors and readers versus the use of other methods of sharing, has not been demonstrated. On the other hand, simply sharing the code underlying a publication in a repository has been shown to bring benefits to authors, such as acting as a signal of credibility (McKiernan et al., 2016) and increased citations of the article (Vandewalle, 2012), which has similarly been shown for data sharing (Piwowar, Day & Fridsma, 2007; Colavizza et al., 2020).

Whilst quality and reusability of code is very important for increasing the reproducibility, trust and transparency of research; the lack of shared code is still a huge issue that needs to be overcome. Serghiou et al. (2021) found that 70% of publishers have never published an article with shared code when analysing over 2.7 million articles in PubMed Central (PMC), and only 2.5% of published articles share code. PLOS journals have higher code sharing rates, with 41% of PLOS Computational Biology article sharing code in 2019 (Serghiou, 2021).

Additional time to prepare code for sharing

Additional effort is required to produce interactive and executable versions of published research but our survey showed that even for those researchers already engaged in code sharing, the majority (64%) would not be willing to spend more than a day using a tool that makes code easier to read and run. This suggests that the average researcher may be unwilling to incur additional costs (in time, effort or expenditure) themselves to achieve these outputs, supporting a need for different models for funding and producing these outputs—at least until such time as they can be produced more efficiently. Asking people to predict their future behaviour can lead to overestimation of positive effects (Wood et al., 2016) and therefore it is possible that the number of researchers unwilling to spend more than a day on a new tool is actually higher than 64%. During the pilot at Nature journals, the creation of a code capsule took a median time of nine days (Nature Biotechnology, 2019). Time has been found to be a barrier to sharing other research outputs, such as data, in other studies as well (see, amongst others, Perrier, Blondal & MacDonald, 2020; Tenopir et al., 2020; Digital Science et al., 2021)

Given the mixed feelings of researchers regarding features of interactive notebooks that are not related to code access, and the lack of desire to invest the required effort to produce them, PLOS Computational Biology has opted for the time being to focus on policy and guidance rather than technological solutions to improve code sharing. The importance of these cultural solutions is often underestimated in relation to reproducible code (Samota & Davey, 2021). At PLOS Computational Biology, we observed a high degree of voluntary code sharing (Cadwallader et al., 2021) before implementation of a mandatory policy, and preliminary results of the impact of the policy on the amount of code shared look positive in line with what has been learnt from implementing mandatory versus optional but encouraged data sharing policy, with the latter causing little change to the status quo (Christensen et al., 2019; Colavizza et al., 2020; Statham et al., 2020). We are focusing on supporting good foundational behaviours by authors that we know are important, such as sharing code with good documentation and metadata (Kim, Poline & Dumas, 2018; Stodden et al., 2016). As more code associated with publications is made available as a result of these activities, we anticipate there will be more opportunities to understand how the quality, reusability, and interactivity of shared code affect reproducibility—and the role of technological solutions.

Limitations

One possible limitation of this study is non-response bias. As no incentive was offered to complete the survey, respondents who are already motivated to engage with code sharing may have been more likely to participate. The survey was also directed at computational biologists and related disciplines therefore may not be applicable to all disciplines. It is also worth noting that only 34% of respondents identified as working specifically in the computational biology field. Also, there is an uneven distribution in terms of the number of published papers, with most respondents having published fewer than 20 papers, which may limit the generalisability of the findings to other researchers at other career stages. The geographical spread of our respondents also limits the generalisability of our findings. The survey did not give explanations of the different methods of code sharing and assumed the respondents to be familiar with terms such as “code capsule” and “archived in an open access repository”.

Conclusions

The survey findings have given some valuable insights into researcher behaviour and attitudes towards code sharing and more interactive, executable or reproducible publication formats –which require much effort to create. We have observed a “negative result” with regard to clear opportunities for implementing new features and services in the publishing workflow, but we have a better understanding of why researchers look at code—this predominantly seems to be to better understand the article and code used. This is an issue that could be addressed with multiple potential solutions that we did not evaluate, such as reporting guidelines for methods of relevant studies. Further, the results suggest that researchers are on the whole satisfied with code being shared via a code repository, such as GitHub, because this is a well used tool that gives the user freedom to use the code how they wish (e.g., download, fork, read through). Good accompanying documentation is important to researchers and whilst they think their ability to produce documentation is good, the readers of their code may disagree.

Authors of code have variable practices when it comes to the amount of time they spend preparing code. It is unclear if those spending minimal amounts of time preparing code are doing so because their code is already well prepared for sharing, or because they do not attach much importance to spending time preparing their code as it is not regarded as necessary for career advancement, or because they do not have the time to spend on preparation. The NeuroLibre interactive code notebook demonstrated that readers find many of the features valuable and overall they are generally supportive of notebooks but do not see them as revolutionary in the way code is shared. For publishers wishing to experiment with or implement interactive features or versions of articles, it is important to note that researchers (authors) are likely to need additional support or funding to be incentivised to create these outputs. For publishers wishing to increase code sharing, policy may be a more effective solution, in the computational biology community.

The authors thank James Harney, Gary Beardmore, Helen McDonald and Philip Mills from PLOS for their contributions to the survey work. We also thank James Harney, Marcel LaFlamme and Dan Morgan from PLOS and Professor Jason Papin, University of Virginia and PLOS Computational Biology co-Editor-in-Chief, for comments on an earlier version of this manuscript. We would also like to thank NeuroLibre for the creation of the prototype notebooks and engaging in experimentation with us.

Additional Information and Declarations

Competing Interests

Author Contributions

Data Availability

1 GitHub was the most highly named code repository in all areas of the survey. Bitbucket had a small number of mentions by name.

2 Zenodo was the most highly named archive repository in all areas of the survey. OSF was also mentioned in this context.

Both authors are employees of Public Library of Science (PLOS).

Lauren Cadwallader conceived and designed the experiments, performed the experiments, analyzed the data, prepared figures and/or tables, authored or reviewed drafts of the article, and approved the final draft.

Iain Hrynaszkiewicz conceived and designed the experiments, authored or reviewed drafts of the article, and approved the final draft.

The following information was supplied regarding data availability:

The survey instrument used and anonymised survey data is available at Figshare: Cadwallader, Lauren; Hrynaszkiewicz, Iain; Harney, James (2022): Data from: A survey of researchers’ code sharing and reuse practices, and assessment of interactive notebook prototypes. figshare. Dataset. https://doi.org/10.6084/m9.figshare.19122611.v1.

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
