# Peer review of "A survey of researchers’ code sharing and code reuse practices, and assessment of interactive notebook prototypes"

_PeerJ, doi:10.7717/peerj.13933_

## Round 0.1 · original submission · Minor Revisions

Two reviewers mentioned lack of access to original survey raw data. Please provide a DOI to the data or a statement why this data is not accessible upon publication.

·

Basic reporting

figure 2 could benefit of absolute numbers in the bars

I could not identify a link to the raw shared data.

Experimental design

no comment

Validity of the findings

I missed a statement about the availability of the raw survey data to why that cannot be provided

Additional comments

I like your effort and willingness to understand the question of code publication and sharing.

·

Basic reporting

The manuscript is well written, properly structured and easy to understand. Context and previous findings are adequately provided by references.

It would be helpful if the authors share the raw anonymized survey results.

Experimental design

The research questions are clearly defined and the data collection - 188 completed responses to a survey generated for this study - offers the required data to answer them. The data analysis is solid and several figures help to community the core findings efficiently. Table 4 is rather overwhelming and could also be translated into a figure to make it easier to understand the data.

Validity of the findings

The analysis is sound and understandable as well as supported by several references. The authors contextualize the outcome well in the elaborated discussion part. Furthermore, the author stated as employees of PLOS Computational Biology that the journal has made the strategic decision based on the survey results to rather focus on policies than offering further technological solutions to facilitate the sharing of code for their community.

Additional comments

no comment

·

Basic reporting

no comment.
The article is clear and unabigious. There are many well chosen litature references, All of the data, figures and raw data is accessible and well-structured.

Experimental design

188 researchers in computational biology were surveyed. The findings also include other research areas with partly a very low number of responses. The relevance for those areas from the statistics shown is somewhat questionable.

Validity of the findings

no comment

Additional comments

The idea to include the new code notebook initiative, Neurolibre, into the survey was very good. It gave a feedback on how those tools are accepted, may be it takes more time to further increase this acceptance.

---

## Round 0.2 · accepted · Accept

I have not heard from the reviewers, yet. However, since it's holiday time, they may be away. Moreover, only one of them voted "minor changes" and I think you have adequately addressed these concerns. Thus, I am happy to let you know that your manuscript has been accepted.